# PD-1 Immune Checkpoint Blockade Promotes Therapeutic Cancer Vaccine to Eradicate Lung Cancer

**DOI:** 10.3390/vaccines8020317

**Published:** 2020-06-18

**Authors:** Pournima Kadam, Sherven Sharma

**Affiliations:** 1Molecular Gene Medicine Laboratory, Veterans Affairs Greater Los Angeles, Healthcare System, 11301, Wilshire Blvd, Los Angeles, CA 90073, USA; Pournima.Kadam2@va.gov; 2Department of Medicine, UCLA Lung Cancer Research Program, David Geffen School of Medicine at UCLA, Los Angeles, CA 90095, USA; 3Jonsson Comprehensive Cancer Center, David Geffen School of Medicine at UCLA, Los Angeles, CA 90095, USA

**Keywords:** lung cancer, PD-1, immune checkpoint, CCL21, dendritic cells, immunotherapy, tumor lysate antigen

## Abstract

(1) Background: Targeting inhibitory immune checkpoint molecules has highlighted the need to find approaches enabling the activation of immune responses against cancer. Therapeutic vaccination, which induces specific immune responses against tumor antigens (Ags), is an attractive option. (2) Methods: Utilizing a K-RasG12Dp53null murine lung cancer model we determined tumor burden, tumor-infiltrating T cell (TIL) cytolysis, immunohistochemistry, flow cytometry, and CD4 and CD8 depletion to evaluate the efficacy of PD-1 blockade combined with CCL21-DC tumor lysate vaccine. (3) Results: Anti-PD-1 plus CCL21-DC tumor lysate vaccine administered to mice bearing established tumors (150 mm^3^) increased expression of perforin and granzyme B in the tumor microenvironment (TME), increased tumor-infiltrating T cell (TIL) activity, and caused 80% tumor eradication. Mice with treatment-induced tumor eradication developed immunological memory, enabling tumor rejection upon challenge and cancer-recurrence-free survival. The depletion of CD4 or CD8 abrogated the antitumor activity of combined therapy. PD-1 blockade or CCL21-DC tumor lysate vaccine monotherapy reduced tumor burden without tumor eradication. (4) Conclusion: Immune checkpoint blockade promotes the activity of the therapeutic cancer vaccine. PD-1 blockade plus CCL21-DC tumor lysate vaccine therapy could benefit lung cancer patients.

## 1. Introduction

Lung cancer is the leading cause of cancer death among men and women in the United States and worldwide; more than 1.9 million people died from lung cancer in 2017 [1]. Long-term survival after resection for non-small-cell lung cancer (NSCLC) is about 50% with advances in radiation therapy, chemotherapy, and molecular targeted therapies [1]. Targeting inhibitory immune checkpoint molecules and immune-suppressive targets underscores the need for approaches enabling the induction and activation of immune responses against cancer [2]. Therapeutic cancer vaccines, which induce a specific immune response against tumor antigens (Ags) are an attractive option. This strategy has shown low clinical efficacy when combined with other treatment modalities [3]. As recent advances that led to the approval of ipilimumab, PD-1 and PD-L1 inhibitors have revolutionized cancer immunotherapy and become well established as highly effective treatment options for managing patient outcomes [4,5]. However, not all patients respond to these treatments [6]. Although therapeutic cancer vaccines can induce a specific T cell immune response against tumor Ags, the PD-1/PDL-1 T cell immune evasion pathway can downregulate a more robust response [1]. Therapeutic cancer vaccines combined with PD-1 immune checkpoint blockade therapy is a reasonable approach with the potential for cancer-free survival [7]. We hypothesize that PD-1 immune checkpoint blockade will increase the amplitude of therapeutic cancer vaccine mediated activated T cell responses in the tumor microenvironment (TME). In this study, we combined dendritic cells (DCs) with tumor Ags and CCL21 to enhance T cell infiltration of tumors. In T cell areas of the secondary lymphoid organ, CCL21 stimulates the recruitment of T cells and Ag-loaded DCs through CCR7 and CXCR3 receptors that progress to T cell activation [8]. CCL21-DC tumor Ag vaccine administered *s.c.* enables repeat dosing to elicit T-cell-dependent and systemic antitumor responses. The goal of this study is to determine the mechanisms of the PD-1 blockade and the therapeutic cancer vaccine in the modulation of immune activity in K-RasG12Dp53null lung adenocarcinoma, which is common in smokers and non-smokers. We hypothesize that the CCL21-DC tumor Ag vaccine mediated specific systemic immunity will benefit patients who have limited CD8 T cell infiltration of their tumors and or limited expression of tumor PD-L1. We report that PD-1 immune checkpoint blockade combined with CCL21-DC tumor Ag vaccine eradicates tumors and has the potential to augment therapy in lung cancer patients who have low baseline tumor T cell infiltration and do not respond to PD-1 therapy.

## 2. Materials & Methods

### 2.1. Cell Line and Reagents

The murine K-RasG12Dp53null lung adenocarcinoma was isolated from lung tumors of the K-RasG12Dp53null transgenic mice obtained from J. Kurie (MD Anderson Houston, TX, USA). The culture medium (CM) was RPMI 1640 supplemented with 10% fetal bovine serum (Gemini Bioproducts West Sacramento, CA, USA), 100 units/mL penicillin, 0.1 mg/mL streptomycin, and 2 mM glutamine (JRH Biosciences Lenex, MA, USA). Fluorescein isothiocyanate-, phycoerythrin-, allophycocyanin-, PerCP-, or APC-Cy7-conjugated anti-mouse Abs to CD4 (RM4-5) and CD8a (53-6.7) were from BD Biosciences or eBiosciences. For in vivo experiments, anti-PD-1 (BE0146), anti-CD4 (L3T4), and anti-CD8 (YTS169.4) were from BioXCell. Isotype control antibody (Ab) was from Sigma. The Abs used for T cell depletion were different for monitoring T cell levels. Bradford protein quantification dye was obtained from Sigma. Tissue digestion buffer was obtained from Miltenyi and used in Miltenyi tissue homogenizer according to the manufacturer’s instructions. T cell purification columns were from R&D.

### 2.2. Cell Culture

K-RasG12Dp53null lung cancer cells and bone-marrow-derived DCs from femurs of syngeneic 129-E mice were routinely cultured in Corning T75 tissue culture flask in humidified atmosphere containing 5% CO_2_ in air in CM. K-RasG12Dp53null and DCs were mycoplasma and murine viral pathogen free. The K-RasG12Dp53null cell line was used up to the 10th passage. DCs were cultured for 7 days in GM-CSF (20 ng/mL) and IL-4 (4 ng/mL) containing CM. The DCs were plated in flasks coated with 2% gelatin (Sigma). Non-adherent DCs were harvested on day 7 for experiments. The cultured DCs had increased levels (70–90%) of MHC class I and MHC class II expression, as evaluated by staining/flow cytometry (data not shown).

### 2.3. CCL21 Transduction and Tumor Lysate Pulsing of DCs

DC were transduced with a replication-deficient adenovirus expressing murine CCL21 or control virus without CCL21 insert at a MOI of 100:1 and centrifugation at 2000× *g* for 2 h. CCL21-transduced DCs produced 10–16 ng CCL21/10^6^ cells/24 h, as evaluated by CCL21-specific ELISA. The transduced DCs were pulsed with K-RasG12Dp53null tumor lysates prepared by digesting K-RasG12Dp53null tumors (day 15–20) with a Miltenyi Tissue dissociation kit in the gentleMACS Octo Dissociator at 37 °C for 40 min. Following digestion, tumor cells were heated for 5 h at 42 °C and recovered at 37 °C for 24 h in CM containing 10% mouse serum (MS). The cells were washed three times with PBS and suspended in RPMI followed by disruption by five freeze (liquid nitrogen) and thaw (37 °C) cycles. After freeze–thaw the lysates were passed through a 28-gauge needle 10 times. Tumor lysates were filtered through a 0.2 µm syringe filter. The protein content was determined by Bradford assay, and aliquots (0.5 mg/aliquot) were stored at −80 °C. Lysates were tested for bacterial endotoxin with the Limulus Amoebocyte lysate assay (Lonza, Alpharetta, USA) according to instructions, and preparations routinely contained less than 0.01 EU/µg protein. DCs (50 × 10^6^) were pulsed with tumor lysates (200 µg/mL) for 24 h in RPMI. Then, a 5 × 10^6^ CCL21-DC tumor Ag vaccine/injection was utilized. The cells were washed three times with PBS and reconstituted in sterile PBS for injection.

### 2.4. Tumorigenesis

Tumorigenesis was conducted to test whether anti-PD-1 would augment CCL21-DC tumor lysate vaccine over monotherapy. Here, 6–8-week-old pathogen-free 129-E mice (Charles River Lab) were used in the experiments, and tumor volume was determined using the following formula: Tumor Volume = 0.4ab^2^, where a = large diameter and b = small diameter. K-RasG12Dp53null tumor cells (10^6^) were injected on right supra scapular region of 129-E mice. The mice with 150 mm^3^ tumors, were injected with CCL21-DC lysate Ag pulsed vaccine (5 × 10^6^) once per week for 3 weeks by *s.c.* administration on the left contra-lateral flank of the tumor. Mice were injected *i.p.* with anti-PD-1 (200 µg/dose) or isotype IgG2b Ab (200 µg/dose) every 48 h for 3 weeks. The anti-PD-1 was non-depleting neutralizing Ab. To determine the induction of immunological memory, mice that eradicated tumors in response to therapy were challenged by *s.c.* injection on the left flank with 10^6^ K-RasG12Dp53null tumor cells. One week following the 2nd week of therapy, tumors were harvested to conduct the experiments described below. All animal work was conducted in accord with the Veterans Affairs Institutional Animal Care and Use Committee guidelines (id D16-00002). The Animal Care and Use Committee review board approved all the studies involving animals. Male and female mice were used in same proportion in all the experiments. Mice were re-challenged one month following tumor eradication with 10^6^ tumor cells. The depleting anti-CD4 (L3T4) and anti-CD8 (YTS169.4) were from BioXCell. The monitoring anti-CD4 monoclonal antibody (GK1.5) was from eBioscience. The monitoring anti-CD8 monoclonal antibody (5H10) was from Invitrogen.

### 2.5. Flow Cytometry

Tumors were digested in the gentleMACS Octo Dissociator for evaluation by specific staining/flow cytometry. Flow cytometry analysis was conducted to determine CD8 T cells (FITC conjugated anti- mouse CD8) expressing granzyme B (PE conjugated anti-mouse granzyme B) in the TME following diluent, anti-PD-1, CCL21-DC lysate vaccine, and anti-PD-1 plus CCL21-DC lysate vaccine therapy. Fluorimetry staining followed by flow cytometry analyses were performed on single-cell suspensions of tumor digest. A total of 10,000 events were acquired on the Cytoflex flow cytometer (Beckman Coulter, Inc, Carlsbad, USA), and the data were analyzed by CytExpert software. Cell staining was performed with fluorochrome-labeled specific Ab on 10^6^ single-cell suspension of tumor digest according to manufacturer’s instructions.

### 2.6. H&E

H&E staining was conducted on tumor sections.

### 2.7. Immunohistochemistry (IHC) Staining

IHC staining was conducted on tumor sections. CD3 T cells, apoptotic tumor cells, perforin, and granzyme B were assessed by microscopic examination of IHC-stained sections (brown) with a calibrated graticule (a 1-cm^2^ grid subdivided into one hundred 1-mm^2^ squares). A grid square with specific staining occupying >50% of its area was scored as positive and the total number of positive squares was determined. Ten separate fields from four IHC sections from four mice per group were examined under high power (20× objective)

### 2.8. CD4T-Cell and CD8T-Cell Depletion

Twelve-day tumor-bearing mice were individually treated with the respective T effector cell depleting Abs (200 µg/dose) or isotype control via *i.p.* injection every 48 h for the duration of the experiment. This schedule depleted the respective T effector cell as determined by staining/flow cytometry analyses of TME. Following CD4 and CD8 depletion, there was >95% depletion of the respective T effector cells (Appendix A). Abs used to monitor T cells were different from the depletion Abs.

### 2.9. In Vitro Cytolysis

Tumor-infiltrating T cells (TILs) from 12-day treatment groups were purified by Miltenyi beads and co-cultured with tumor cells in CM at a ratio of 5:1 for 24 h. Almar blue (20 µL) was added for 4 h, and the fluorescence was read at excitation/emission (530nm/595nm) in a Wallac Fluorescence reader.

### 2.10. DC Activity in TME Following Anti-PD-1 Plus CCL21-DC Tumor Lysate Vaccine Therapy

CD4 and CD8 T cells were purified with Miltenyi beads from the spleens of mice one month following vaccination (once per week for 3 weeks) with DC tumor lysate vaccine. Purified DCs from the TME of (i) diluent control, (ii) anti-PD-1, (iii) CCL21-DC tumor lysate vaccine, and (iv) anti-PD-1 plus CCL21-DC tumor lysate vaccine were pulsed with MHC Class I K-Ras peptide (LVVVGADGV) or MHC Class II (MTEYKLVVVGADGVG) and co-cultured with splenic CD8 or CD4 T cells of the vaccinated mice at a ratio of 1:5 for 24 h. IFNγ secreted by T cells in the co-culture was determined by IFNγ-specific ELISA (eBioscience).

### 2.11. Statistical Analyses

All data are presented as mean ± SE. Statistical analysis was performed using Prism (GraphPad Software, San Diego, USA). We used analysis of variance for data with multiple groups, and we used unpaired Student’s *t*-test for dual comparison. *p*-values < 0.05 were considered significant.

## 3. Results

### 3.1. Anti-PD-1 Augments CCL21-DC Tumor Ag Vaccine Antitumor Activity

We tested the antitumor efficacy of (1) diluent, (2) anti-PD-1, (3) CCL21-DC tumor lysate vaccine, and (4) CCL21-DC tumor lysate vaccine plus anti-PD-1 in the K-RasG12Dp53null tumor model. The treatments induced the following fold reductions in tumor burden compared to diluent control: (1) anti-PD-1 induced a 2-fold reduction, (2) CCL21 DC tumor lysate vaccine induced a 2-fold reduction, and (3) Anti-PD-1 plus CCL21-DC tumor lysate vaccine induced a 17-fold reduction. The anti-PD-1, CCL21-DC tumor lysate vaccine, and anti-PD-1 plus CCL21-DC tumor lysate vaccine treatments resulted in 3-, 3-, and 19-fold weight changes of tumors at the end of therapy in comparison to control, respectively. In comparison to the monotherapy that reduced tumor burden without causing tumor eradication, treatment groups receiving CCL21-DC tumor lysate vaccine plus anti-PD-1 led to 80% tumor eradication (Table 1). The cured mice rejected a secondary tumor challenge and remained tumor-free, demonstrating long-term immunological memory. Individual depletion of CD4T cells or CD8T cells (Figure 1A) abrogated the antitumor activity of combined therapy. Compared to diluent control, combined therapy caused a 19-fold tumor weight reduction, whereas that of monotherapy was 3-fold (Figure 1C). DC lysate and anti-PD-1 plus control vector (CV)-DC lysate caused 10% tumor eradication (data not shown). This result indicates that CCL21 secretion by DCs pulsed with tumor Ags enhanced tumor eradication by 8-fold in comparison to DCs or anti-PD-1 plus CV-DC tumor lysate treatment groups.

### 3.2. Combined Therapy Enhances CD8 T Cells Expressing Granzyme B and TIL Cytolysis

T cells from the TME of the combined therapy had increased cytolytic activity (8-fold) against parental K-RasG12Dp53null tumor cells in vitro in comparison to diluent control, whereas monotherapy had a 2-fold increase (Figure 2A). Flow cytometry of Tregs and NK cells was not conducted because there were no changes in the activity of these cells. Purified Tregs from the TME did not alter the proliferation of anti-CD3/anti-CD28 stimulated T cells. Purified NK cells from the TME did not alter the cytolysis of tumor cells in vitro. Flow cytometry analyses were conducted to evaluate CD8 T cells expressing granzyme B following therapy. Flow cytometry analyses revealed increased frequency (4-fold) of activated CD8 T cells expressing granzyme B in the anti-PD-1 plus CCL21-DC tumor lysate vaccine treatment group in comparison to monotherapy (Figure 2B).

### 3.3. Anti-PD-1 Plus CCL21-DC Tumor Ag Vaccine Therapy Enhances Immune Infiltrates and Caspase 3

We conducted H&E and IHC staining to evaluate tumor immune infiltrates following therapy. In comparison to controls and monotherapy, H&E staining of tumor sections revealed enhanced immune infiltrates with reduced tumor burden in the anti-PD-1 plus CCL21-DC tumor lysate vaccine treatment group (Figure 2A). IHC of the tumor sections revealed an increase in CD3T cells, caspase 3 apoptotic tumor cells, and perforin and granzyme B expression in the anti-PD-1 plus CCL21-DC tumor lysate vaccine treatment group compared to diluent control and monotherapy (Figure 3A). Quantification of IHC-stained paraffin-embedded sections revealed that anti-PD-1 plus CCL21-DC lysate vaccine treatment led to the highest CD3 T cell (5- and 2-fold), perforin (4- and 3-fold), and granzyme B (4-and 3-fold) increase and greatest reduction in tumor burden denoted by enhanced caspase-3-stained apoptotic tumor cells (5-and 3-fold) compared with anti-PD-1 and CCL21-DC lysate vaccine monotherapy (Figure 3B).

### 3.4. Anti-PD-1 Plus CCL21-DC Tumor Ag Vaccine Therapy Enhances DC Activity in the TME

CD4 and CD8 T cells from the spleens of mice were purified with Miltenyi beads following vaccination with DC tumor lysate vaccine. Purified DCs from the TME of (i) diluent control, (ii) anti-PD-1, (iii) CCL21-DC tumor lysate vaccine, and (iv) anti-PD-1 plus CCL21-DC tumor lysate vaccine were pulsed with MHC Class I K-Ras peptide (LVVVGADGV) or MHC Class II (MTEYKLVVVGADGVG) and co-cultured with splenic CD8 or CD4 T cells of the vaccinated mice at a ratio of 1:5 for 24 h. IFNγ secreted by T cells in the co-culture was determined by IFNγ-specific ELISA. In comparison to diluent control or monotherapy, anti-PD-1 plus CCL21-DC tumor lysate Ag induced the highest DC activity of presenting MHC Class I (27-fold) and MHC Class II (36-fold) K-RasG12D tumor peptides to tumor-vaccine-specific T cells. DC activity following monotherapy with anti-PD-1 presented MHC Class I (4-fold) and MHC Class II (4-fold) to tumor vaccine specific T cells. DC activity following monotherapy with CCL21-DC tumor lysate presented MHC Class I (3-fold) and MHC Class II (3-fold) to tumor vaccine specific T cells. Control peptide (FECNTAQAC)-pulsed DCs did not stimulate T cell IFNγ production (Figure 4).

Purified NK cells were plated with tumor cells at the ratio of 5:1 for 24 h. Almar blue (20 µL) was added for 4 h after 24 h, and the fluorescence was read. Bars represent SE. Results are representative of a single experiment (Figure 5B) (*n* = 4 mice/group).

Anti-CD4 and anti-CD8 led to >95% depletion of the respective T cell populations. Results are representative of two experiments (*n* = 10 mice/group).

## 4. Conclusions

Targeting inhibitory immune checkpoint molecules has highlighted the need to find approaches enabling the activation of immune responses against cancer. Therapeutic vaccination, which induces specific immune responses against tumor Ags, is an attractive option. Although this strategy has shown minimum clinical efficacy when combined with other treatment modalities, therapeutic cancer vaccine strategies have a high probability of success when combined with immune checkpoint blockade therapy with the potential for cancer-free survival. This could be achieved by evaluating therapeutic vaccines combined with immune checkpoint blockade protocols in preclinical models prior to clinical translation. As effective therapeutic vaccines and combinations with biological therapy are defined in preclinical models, the understanding of the interactions in the TME that modulate antitumor activities can be utilized to develop effective technologies for long-term cancer-free survival.

## 5. Discussion

We evaluated PD-1 blockade combined with CCL21-DC tumor Ag pulsed vaccine in the K-RasG12Dp53null murine model of lung adenocarcinoma. We are currently conducting a phase I clinical trial of CCL21-DC administered intratumorally in patients with advanced-stage lung cancer [9]. This approach utilizes an intratumoral administration of CCL21-DC that is limited by the geographical location of the tumor and the number of times the tumor can be injected. While this trial is ongoing in the right clinical setting, we are evaluating the systemic CCL21-DC tumor lysate vaccine. The systemic therapeutic vaccine allows for multiple noninvasive *s.c.* injections. DCs are highly specialized professional APCs with the capacity to capture, process, and present Ags to T cells, resulting in T cell activation. In the current study, the data demonstrate that PD-1 blockade augments T cell and DC activity of CCL21-DC tumor Ag vaccine in the TME to eradicate K-RasG12Dp53null lung adenocarcinoma. This suggests that a combination of PD-1 blockade therapy with therapeutic vaccination could prove essential to improve the management and clinical outcome of patients who do not respond or for those whose disease eventually progresses.

The results of this study present a unique translational opportunity because they provide a means for the activation of T cells to effectively target and eradicate the aggressive K-RasG12Dp53null mutant lung cancer. p53 normally serves as a tumor suppressor, the function of which is lost upon allelic loss or mutation. Lung cancers have a high p53-specific mutation in adenocarcinoma and in squamous cells. In the K-RasG12Dp53null tumor model, tumors develop in an immune-competent host. Our results demonstrate that this model responds to individual therapeutic vaccination or immune checkpoint blockade therapy, although the therapies administered individually are sub-optimal. This model has low baseline activated T cell infiltrates, similar to NSCLC patients who have low responses to PD-1 blockade. [6,10]. The K-RasG12Dp53null model is representative of a significant mutational phenotype of NSCLC. Tobacco mutagens cause a high mutation frequency in the somatic genomes of smoking-associated tumors. Our data demonstrate that *s.c.* administration of CCL21-DC tumor lysate vaccine results in tumor growth inhibition, but the vaccine therapy alone is not optimal. Immunomodulatory monoclonal Abs that target immune evasion pathways in cancer have shown efficacy in clinical trials [10,11,12]. The K-RasG12Dp53null model does not have the high mutation load seen in human lung cancer. PD-1 immune checkpoint blockade therapy is more effective in tumors harboring a high mutation burden [13,14]. T cells respond broadly to mutation changes that render proteins immunogenic, and the high mutation load provides additional targets for T cells that may increase the efficacy of response. The high mutation load in smoking-induced lung cancer provides an opportunity for developing multiepitope-based personalized vaccines. T cells also respond to driver oncogenic proteins and aberrant protein expression caused by the driver mutations and loss of tumor suppressor genes. However, since no single mouse cancer model recapitulates all the key aspects of human disease, evaluation of CCL21-DC tumor Ag vaccine in combination with PD-1 blockade therapy in several lung cancer models will provide information on the general efficacy of the approach. Non-smokers who do not have a high mutation burden but develop lung cancer may also benefit from this combined approach. We found that the CCL21-DC tumor Ag vaccine in combination with PD-1 blockade therapy rescued TIL activity, causing enhanced tumor cytolysis. Tumor eradication was T-cell-dependent because the depletion of T cells abrogated the effect. CCL21-DC tumor Ag vaccine plus anti-PD-1 induced the highest expression of perforin and granzyme B expression in the TME compared to monotherapy. PD-1/PD-L1-mediated immune suppression promotes tumor progression in the K-RasG12Dp53null model, suggesting that the success of immune therapy is dependent on the inhibition of this mechanism.

Currently, up to 50–60% of patients being treated with PD-1 checkpoint inhibitors do not have T cell infiltration and have reduced antitumor response. Our data demonstrate that the CCL21-DC tumor Ag vaccine has the means to overcome this limitation for long-term cancer-free survival and provide justification for evaluation for clinical translation.

## Figures and Tables

**Figure 1 vaccines-08-00317-f001:**
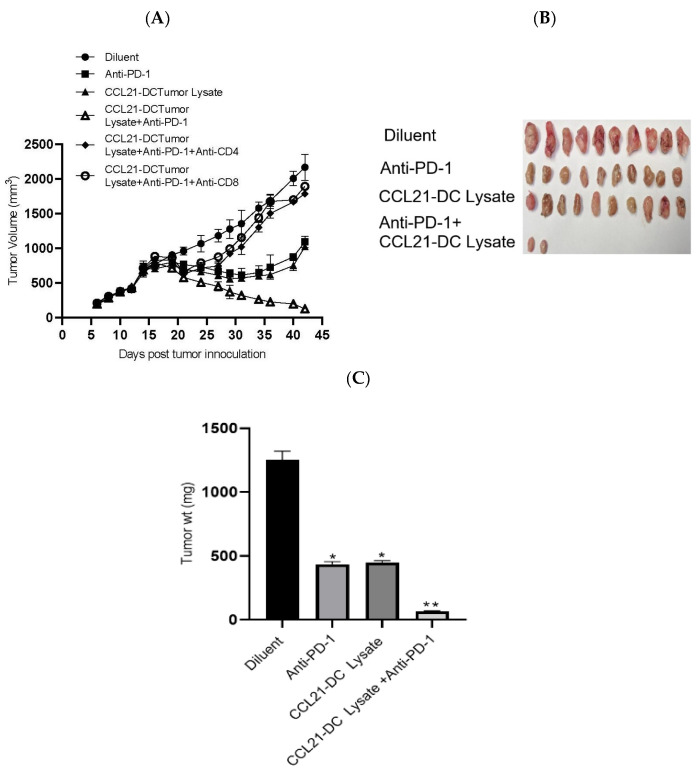
K-RasG12Dp53null tumor cells (10^6^) were inoculated in the supra scapular region of 129-E mice. Mice bearing 12-day established tumors were treated with (**i**) diluent, (**ii**) anti-PD-1, (**iii**) CCL21-DC lysate vaccine, (**iv**) CCL21-DC lysate vaccine plus anti-PD-1, (**v**) CCL21-DC lysate vaccine plus anti-PD-1 plus anti-CD4, and (**vi**) CCL21-DC lysate vaccine plus anti-PD-1 plus anti-CD8. In comparison to monotherapy, combined therapy was more effective at inhibiting tumor growth (**A**). Depletion of CD4 T or CD8 T cells abrogated antitumor activity of combined therapy (**A**). Combined therapy reduced the weight of tumors in comparison to monotherapy and control (**B**,**C**); ** *p* < 0.01 in comparison to diluent control, * *p* < 0.05 in comparison to monotherapy. Results are representative of an independent experiment. The experiment was repeated twice (*n* = 10 mice/group).

**Figure 2 vaccines-08-00317-f002:**
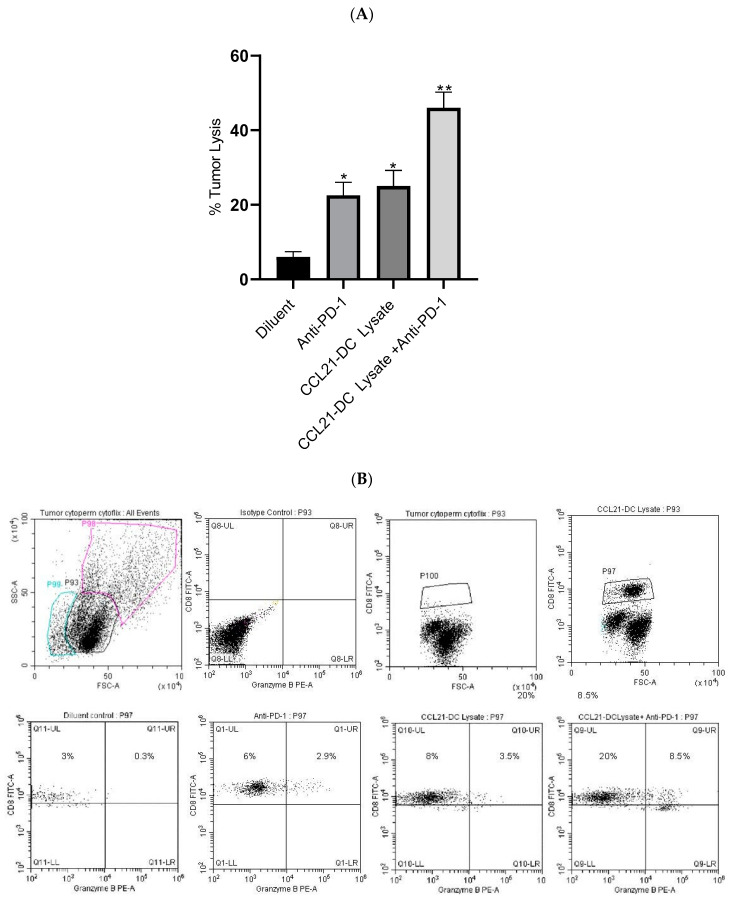
Tumor-infiltrating T cells (TILs) were incubated with K-RasG12Dp53null tumor cells at ratio of 5:1 overnight followed by addition of almar blue for 4 h. TILs from mice treated with combined therapy were most effective at lysing K-RasG12Dp53null tumor cells; ** *p* < 0.01 in comparison to diluent control, * *p* < 0.05 compared to monotherapy. Results are representative of an independent experiment (**A**). The experiment was repeated twice (*n* = 4 mice/group). Flow cytometric analyses of single-cell suspensions of the tumor microenvironment (TME) following therapy showed enhancement of CD8 T cells expressing granzyme B (*p* < 0.05) compared to monotherapy. P98 is dumped cell population, P99 is dead cell population and P100 is CD8 T cell population. Results are representative of an independent experiment. The experiment was repeated twice (*n* = 4 mice/group) (**B**).

**Figure 3 vaccines-08-00317-f003:**
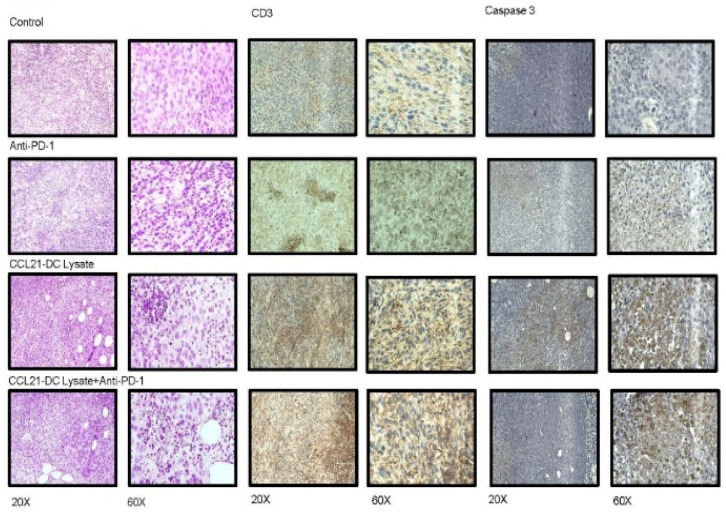
H&E revealed enhanced immune infiltrates and reduced tumor staining following combined therapy. (**A**) IHC of tumor sections revealed enhanced (i) CD3 T cell staining (brown), (ii) tumor cell apoptosis (brown), (iii) perforin, (iv) and granzyme B staining (brown) following combined therapy. (**B**) Stained areas of tumors were quantified by microscopy of IHC-stained paraffin-embedded sections. Anti-PD-1 plus CCL21-DC lysate vaccine treatment led to the highest levels of CD3 T cells, perforin, and granzyme B and greatest reduction in tumor burden, as denoted by enhanced caspase-3-stained apoptotic tumor cells compared with diluent control and monotherapy. Bars represent SE; *p* < 0.01 compared with diluent-treated control; *p* < 0.05 compared with monotherapy. Results are representative of a single experiment (*n* = 4 mice/group).

**Figure 4 vaccines-08-00317-f004:**
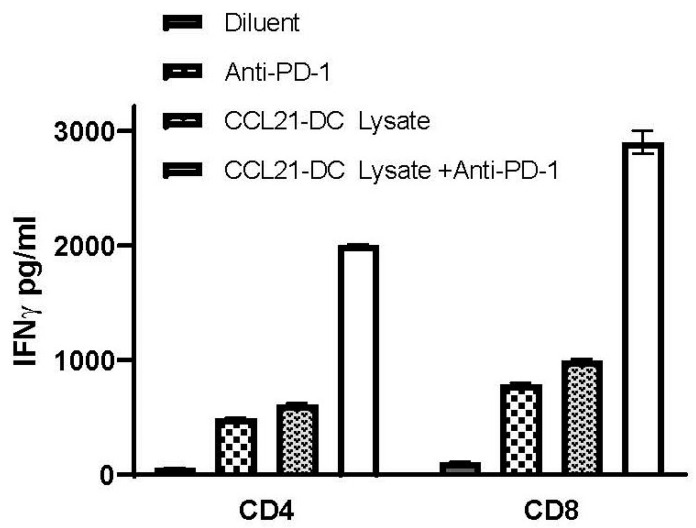
Anti-PD-1 plus CCL21-DC tumor antigen (Ag) vaccine therapy enhances dendritic cell (DC) activity in the TME. CD4 and CD8 T cells were purified from the spleens of mice following vaccination with DC tumor lysate vaccine. DCs from TME of (i) diluent control, (ii) anti-PD-1, (iii) CCL21-DC tumor lysate vaccine, and (iv) anti-PD-1 plus CCL21-DC tumor lysate vaccine were pulsed with MHC Class I K-Ras peptide (LVVVGADGV) or MHC Class II (MTEYKLVVVGADGVG) and co-cultured with splenic CD8 or CD4 T cells of the vaccinated mice at a ratio of 1:5 for 24 h. IFNγ secreted by T cells in the co-culture was determined by IFNγ-specific ELISA. In comparison to diluent control or monotherapy, anti-PD-1 plus CCL21-DC tumor lysate Ag induced the highest DC activity of presenting MHC Class I and MHC Class II K-RasG12D tumor peptides to T cells. Control peptide (FECNTAQAC)-pulsed DCs did not stimulate T cell IFNγ production (data not shown). Bars represent SE; *p* < 0.01 compared with diluent-treated control, *p* < 0.05 compared with monotherapy. Results are representative of a single experiment (*n* = 4 mice/group).

**Figure 5 vaccines-08-00317-f005:**
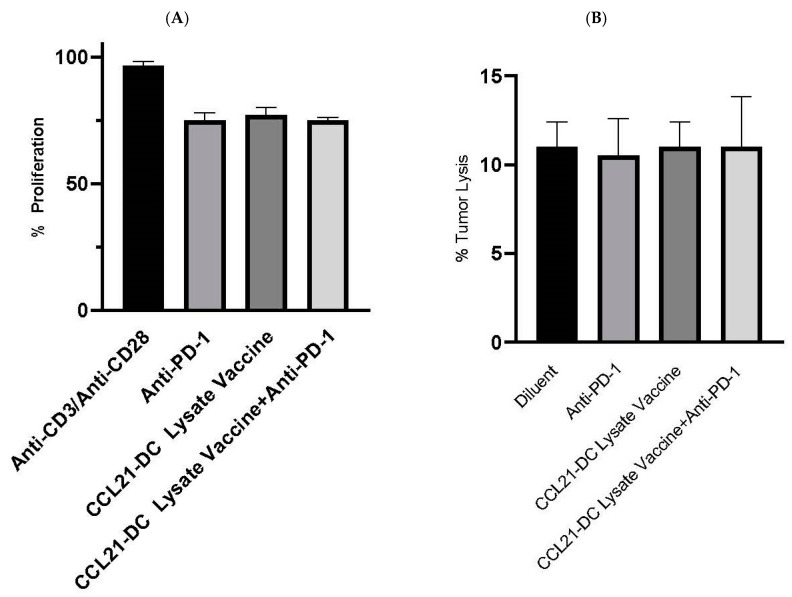
(**A**,**B**) Tregs and NK cells from CCL21-DC lysate vaccine plus anti-PD-1 did not have a change in activity. Purified Tregs (1:5) from combined therapy did not alter the proliferation of anti-CD3/anti-CD28 (0.2/2 µg/mL) stimulated T cell proliferation. Almar blue (20 µL) was added for 4 h on day 3, and fluorescence was read at excitation/emission (530nm/595nm) in a Wallac Fluorescence reader.

**Table 1 vaccines-08-00317-t001:** Anti-PD-1 blockade plus CCL21-DC lysate vaccine eradicated established tumors.

Treatment	Number of Mice with Tumor Eradication	Number of Mice that Rejected Re-Challenge
Diluent	0/10	N/D
Anti-PD-1	0/10	N/D
CCL21-DC Lysate	0/10	N/D
CCL21-DC Lysate + Anti-PD-1	8/10	8/8

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
