# Peer review of "PD-1 Immune Checkpoint Blockade Promotes Therapeutic Cancer Vaccine to Eradicate Lung Cancer"

_vaccines, 2020, doi:10.3390/vaccines8020317_

Round 1

Reviewer 1 Report

The authors present an interesting study on the utility of CCL21-DC tumor lysate vaccine, in combination with PD-1 blockade, against lung cancer. While the tumor eradication studies are very promising; several major issues exist with regards to methodology, data, and interpretation. Some of my concerns include:

  • Since primary cells are transduced; authors should show transduction efficiency. Authors should also clarify at what day of DC differentiation were the cells transduced.
  • It is essential to show the flow cytometry data to demonstrate the efficient T-cell depletion (even if it is in supplementary data section)
  • Authors claim that CCL21 transduction is the most significant contributor to the therapeutic response; yet no data is shown about the differences in infiltration of T cells into the draining lymph nodes. Additionally, the IHC presented here should be of higher magnification (the insets presented in the figure is blurry).
  • Authors should also show, at the minimum, frequency of CD3, CD4, and CD8 T cells in tumors and DLNs using flow cytometry.
  • The flow cytometry data for CD8 and Granzyme is very unconvincing. The populations do not separate out as they should.
  • Since CCL21:CCR7 axis also modulates other immune cell subsets, authors should present additional data on phenotyping of other immune cell subsets.
  • Lastly, the article needs extensive editing and organization of figures/results section.

Author Response

Reviewer 1

Comment 1

  • Since primary cells are transduced; authors should show transduction efficiency. Authors should also clarify at what day of DC differentiation were the cells transduced.
  • Response to comment 1
  • We evaluated the amount of CCL21 secreted by the transduced DC at 24h. CCL21 transduced DC produced 10-16ng CCL21/106 cells /24h evaluated by CCL21 specific DC were cultured for 7 days in GM-CSF (20ng/ml) and IL-4 (4ng/ml) containing culture medium prior to use.

Comment 2

  • It is essential to show the flow cytometry data to demonstrate the efficient T-cell depletion (even if it is in supplementary data section)
  • Response to comment 2
  • Following CD4 and CD8 depletion there was 85-90% depletion of the respective T effector cells.

Comment 3

  • Authors claim that CCL21 transduction is the most significant contributor to the therapeutic response; yet no data is shown about the differences in infiltration of T cells into the draining lymph nodes. Additionally, the IHC presented here should be of higher magnification (the insets presented in the figure is blurry).
  • Response to comment 3
  • The tumor lysate pulsed DC and CV transduced DC induced 10% tumor eradication is the reason why we did not evaluate frequency of activated T cells in the TME for these groups. We now present the 60X images of IHC separately in fig. 3A.
  • Comment 4
  • Authors should also show, at the minimum, frequency of CD3, CD4, and CD8 T cells in tumors and DLNs using flow cytometry.

Response to comment 4

We have shown the area of stained CD3, caspase 3, perforin and granzyme B in the tumor sections following IHC one week after 2nd therapy. The response of mice following therapy is dynamic. We conducted flow cytometry of CD8 T cells expressing granzyme B in the TME one week following the 2nd therapy and have presented this data. We selected this time frame to reveal early changes in the TME that lead to tumor eradication. We did not evaluate the events in the draining lymph node following therapy but have focused on events in the TME as these events lead to changes in tumor burden as shown by our data.

Comment 5

  • The flow cytometry data for CD8 and Granzyme is very unconvincing. The populations do not separate out as they should.

Response to comment 5

We have replaced flow cytometry data for CD8 and Granzyme B. The data is as presented because it was conducted one week following the 2nd therapy. We selected this time frame because it provides information on the early responses that lead to tumor eradication.

Comment 6

  • Since CCL21:CCR7 axis also modulates other immune cell subsets, authors should present additional data on phenotyping of other immune cell subsets.
  • Response to comment 6
  • We have focused to present data on T cells as depletion of these effectors abrogates the anti-tumor activity of anti-PD-1 plus CCL21-DC tumor lysate vaccine. Flow cytometry of Tregs and NK cells was not conducted because there were no changes in the activity of these cells. Miltenyi bead purified Tregs from the TME did not alter the proliferation of anti-CD3/anti-CD28 stimulated T cells. Miltenyi bead purified NK cells from the TME did not alter the cytolysis of tumor cells in-vitro.
  • Comment 7
  • The article needs extensive editing and organization of figures/results section.

Response to comment 7

We have edited the figures and results section. We have included symbol legend for Figure 1A. We have edited figures 1C and 1D as 2A and 2B. We have edited figures 2A and 2B as figures 3A and 3B. We have edited figure 2C as figure 4.

Reviewer 2 Report

The authors present very interesting studies that can actually have a significant impact on improving lung cancer treatment. The work is well designed, has correctly performed experiments, well written introduction and summary. However, the presentation of results that are also interesting should be improved. They were presented by the authors in a hardly understandable way. The section headings in the results should be improved to highlight the information they contain. The description of the results should not resemble the signature of the figure, but introduce the reader to the relevant results and refer only to the figures and tables. Also the description below table 1 should simply be in the text. Figures 1 and 2 should be divided into at least two, because the charts and photos are too small and therefore difficult to read, and the descriptions are very long.

Author Response

Reviewer 2

Comment 1

The section headings in the results should be improved to highlight the information they contain. The description of the results should not resemble the signature of the figure, but introduce the reader to the relevant results and refer only to the figures and tables. Also the description below table 1 should simply be in the text. Figures 1 and 2 should be divided into at least two, because the charts and photos are too small and therefore difficult to read, and the descriptions are very long.

Response to comment 1

We have revised the section titles in the result section and removed these titles from the legends. We have placed the description of table only in the text and removed table I legend. We have divided figure 1 and 2 into two separate figures. We have included symbol legend for Figure 1A. We have edited figures 1C and 1D as 2A and 2B. We have edited figures 2A and 2B as figures 3A and 3B. We have edited figure 2C as figure 4.

Reviewer 3 Report

Kadam and Sharma describe a K-ras murine lung cancer model successfully treated with immunotherapy using checkpoint blockade plus a dendritic cell vaccine.

 Lung cancer is one of the most common and serious type of cancer worldwide. In the past years, immune check point inhibitors have had a revolutionary impact on cancer treatment. However, only a 20-30% of lung cancer patients respond to this therapy. This is due, at least in part, to low levels of PDL1 in tumor cells, the complexity of tumor microenvironment and specific genomic mutations. Because of some features of immunotherapy responsiveness may apply to various tumors, any advance on the field can help to a better understanding of the tumor biology and significantly improve outcomes for patients fighting cancer.

Although there are several lung cancer vaccines platforms, as the ones based on antigen-specific epitopes, the study from Kadam and Sharma provides a new strategy to take into account and, probably, a new hope for lung cancer patients.

Minor changes:

  • Materials and Methods: point 2.4, text format must be revised.
  • Materials and Methods, males and females mice are used in the same proportions in all the in vivo experiments?
  • Materials and Methods: to test immunological memory, what day mice were re-challenged?
  • Lines 168-170, sentence “Mice that exhibited…” should be removed, as is already mentioned two lines above.
  • Figure 1 must be revised: legend from Fig 1A is not present. Fig 1C, the graph related to tumor weight reduction, corresponds to Fig 1B. Consequently, Fig 1C, about % tumor lysis, must be reassigned. Also, plots from Fig 1D could be improved, since a bad compensation is observed.
  • Results: point 3.3, line 210, is related to figure A and B.

Mayor changes:

  • Treatment of NSCLC with CCL21 modified dendritic cells are already being tested in clinical trials in the past and nowadays. I miss those references in discussion and bibliography.
  • Regarding modified DCs, have the authors performed in vitro experiments to check maturation markers, cytokines release…?
  • It could be considered by the authors the tumor immunophenotyping from vaccinated mice, by testing by flow cytometry different markers for Treg cells, NK, activation T lymphocytes (TIGIT, ICOS, CD44…). Those data could enrich substantially the message of the paper.

Author Response

Reviewer 3

Comment 1

  • Materials and Methods: point 2.4, text format must be revised.

Response to comment 1

We have reformatted the text format in point 2.4 as palatino Linot font size 10.

Comment 2

  • Materials and Methods, males and females mice are used in the same proportions in all the in vivo experiments?

Response to comment 2

Male and female mice were used in same proportion in all the experiments. We have added this to the tumorigenesis section.

Comment 3

  • Materials and Methods: to test immunological memory, what day mice were re-challenged?

Response to comment 3

sharma

Comment 4

Lines 168-170, sentence “Mice that exhibited…” should be removed, as is already mentioned two lines above.

Response to comment 4

Lines 168-170, sentence “Mice that exhibited…” has been removed.

Comment 5

Figure 1 must be revised: legend from Fig 1A is not present. Fig 1C, the graph related to tumor weight reduction, corresponds to Fig 1B. Consequently, Fig 1C, about % tumor lysis, must be reassigned. Also, plots from Fig 1D could be improved, since a bad compensation is observed.

Response to comment 5

We have edited the figure and result section. We have included symbol legend for Figure 1A. We have edited figures 1C and 1D as 2A and 2B. We have edited figures 2A and 2B as figures 3A and 3B. We have edited figure 2C as figure 4. We have replaced flow cytometry data for CD8 and Granzyme B. The data is as presented because it was conducted one week following the 2nd therapy. We selected this time frame because it provides information on the early responses that lead to tumor eradication.

Comment 6

Results: point 3.3, line 210, is related to figure A and B.

Response to comment 6

Line 210 has been changed to “Anti-PD-1 plus CCL21-DC Tumor Ag Vaccine Therapy Enhance Immune Infiltrates and Caspase 3.”

Comment 7

Treatment of NSCLC with CCL21 modified dendritic cells are already being tested in clinical trials in the past and nowadays. I miss those references in discussion and bibliography.

Response to comment 7

We added reference for NSCLC treated with CCL21 modified dendritic cells in clinical trials (Lee JM, Lee MH, Garon E, Goldman JW, Salehi-Rad R, Baratelli FE, et al. Phase I Trial of Intratumoral Injection of CCL21 Gene-Modified Dendritic Cells in Lung Cancer Elicits Tumor-Specific Immune Responses and CD8(+) T-cell Infiltration. Clin Cancer Res 2017;23(16):4556-68 doi 10.1158/1078-0432.Ccr-16-2821). This approach utilizes intratumoral administration of CCL21-DC that is limited by the geographical location of the tumor and the number of times the tumor can be injected. While this trial is on going in right clinical setting we are evaluating systemic CCL21-DC tumor lysate vaccine that allows for multiple non-invasive administrations. 

Comment 8

Regarding modified DCs, have the authors performed in vitro experiments to check maturation markers, cytokines release…?

Response to comment 8

We did not evaluate DC maturation markers but evaluated TME DC presentation of K-Ras tumor peptide Ags to T cells from vaccinated mice as depicted in fig. 4. We measured CCL21 relaese following transduction. CCL21 transduced DC produced 10-16ng CCL21/106 cells /24h evaluated by CCL21 specific ELISA.

Comment 9

It could be considered by the authors the tumor immunophenotyping from vaccinated mice, by testing by flow cytometry different markers for Treg cells, NK, activation T lymphocytes (TIGIT, ICOS, CD44…). Those data could enrich substantially the message of the paper.

Response to comment 9

  • Flow cytometry of Tregs and NK cells was not conducted because there were no changes in the activity of these cells. Miltenyi bead purified Tregs from the TME did not alter the proliferation of anti-CD3/anti-CD28 stimulated T cells. Miltenyi bead purified NK cells from the TME did not alter the cytolysis of tumor cells in-vitro. We have quantified the T cells activation marker granzyme B one week following therapy as depicted in fig. 2B.

Round 2

Reviewer 1 Report

Authors have addressed several of my comments. There are however several weaknesses that still need to be addressed:

  1. For the T-cell depletion percentages, it will be prudent to show a representative dot plot (even if in supplementary section)
  2. In the method section, authors should list the exact clones and catalog/vendor of the depleting antibodies used.
  3. Although authors presented 60X images, the images themselves are too small. There is lot of space in the page, the figures should be enlarged to the point the whole page is covered and details of pictures are visible. This way, the readers can actually see the IHC pictures for themselves clearly. The picture size at the moment beats the purpose of including higher magnification.
  4. Authors mention that significant tumor reduction/eradication was the reason T-cell phenotypic analysis could not be performed. The authors should still be able to analyze the DLNs.
  5. For the granzyme flow cytometry data, my concern was not about the robustness of response; but more about the technical issue of scattering of the populations in the figures presented. I still have doubt about this data; but at the very least, the authors should present complete gating strategy for the granzyme data, including any live/dead/dump gate, CD3, CD4, CD8 etc. that was used prior to gating for granzyme. Authors may chose to provide this in supplementary.
  6. Authors mention no change in ability of Tregs and NK cells to suppress T-cell proliferation. This is a very important data and has implication for this therapy. This data figure should be included in the main text.

Author Response

Response to Reviewer

We thank the reviewer for his comments. We have revised the manuscript accordingly.

Comment 1

  1. For the T-cell depletion percentages, it will be prudent to show a representative dot plot (even if in supplementary section)

Response to Comment 1

We have added dot plot figure S1 to show T cell depletion.

Comment 2

  1. In the method section, authors should list the exact clones and catalog/vendor of the depleting antibodies used.

Response to Comment 2

We have listed the clones/catalog # and vendor of the depleting antibodies in the methods section. Anti-CD4 (L3T4) and anti-CD8 (YTS169.4) were from BioXCell. The monitoring anti-CD4 monoclonal antibody (GK1.5) was from eBioscience.  The monitoring anti-CD8 monoclonal antibody (5H10) was from Invitrogen.

Comment 3

  1. Although authors presented 60X images, the images themselves are too small. There is lot of space in the page, the figures should be enlarged to the point the whole page is covered and details of pictures are visible. This way, the readers can actually see the IHC pictures for themselves clearly. The picture size at the moment beats the purpose of including higher magnification.

Response to Comment 3

We have increased the size of the images.

Comment 4

Authors mention that significant tumor reduction/eradication was the reason T-cell phenotypic analysis could not be performed. The authors should still be able to analyze the DLNs

Response to Comment 4

We did not perform experiments on DLN and that is why no DLN analysis was shown.

Comment 5

  1. For the granzyme flow cytometry data, my concern was not about the robustness of response; but more about the technical issue of scattering of the populations in the figures presented. I still have doubt about this data; but at the very least, the authors should present complete gating strategy for the granzyme data, including any live/dead/dump gate, CD3, CD4, CD8 etc. that was used prior to gating for granzyme. Authors may chose to provide this in supplementary.

Response to Comment 5

The gating strategy prior to CD8 and granzyme B stains is presented.

Comment 6

  1. Authors mention no change in ability of Tregs and NK cells to suppress T-cell proliferation. This is a very important data and has implication for this therapy. This data figure should be included in the main text.

Response to Comment 6

We present data to show Tregs from combined therapy do not alter T cell proliferation and NK cells do not alter tumor cytolysis.   

Round 3

Reviewer 1 Report

The authors have addressed my major concerns. One suggestion is that authors do not add figure numbers to the section titles in the results section.